# Immune-Mediated Drug-Induced Liver Injury: Immunogenetics and Experimental Models

**DOI:** 10.3390/ijms22094557

**Published:** 2021-04-27

**Authors:** Alessio Gerussi, Ambra Natalini, Fabrizio Antonangeli, Clara Mancuso, Elisa Agostinetto, Donatella Barisani, Francesca Di Rosa, Raul Andrade, Pietro Invernizzi

**Affiliations:** 1Centre for Autoimmune Liver Diseases, Division of Gastroenterology, Department of Medicine and Surgery, University of Milano-Bicocca, 20900 Monza, Italy; clara.mancuso@unimib.it (C.M.); donatella.barisani@unimib.it (D.B.); pietro.invernizzi@unimib.it (P.I.); 2European Reference Network on Hepatological Diseases (ERN RARE-LIVER), San Gerardo Hospital, 20900 Monza, Italy; 3Institute of Molecular Biology and Pathology (IBPM), National Research Council of Italy (CNR), 00185 Rome, Italy; ambra.natalini@ibpm.cnr.it (A.N.); fabrizio.antonangeli@cnr.it (F.A.); francesca.dirosa@cnr.it (F.D.R.); 4Academic Trials Promoting Team, Institut Jules Bordet, L’Universite’ Libre de Bruxelles (ULB), 1050 Brussels, Belgium; elisa.agostinetto@humanitas.it; 5Medical Oncology and Hematology Unit, Humanitas Clinical and Research Center—IRCCS, Humanitas Cancer Center, Rozzano, 20089 Milan, Italy; 6Department of Biomedical Sciences, Humanitas University, via Rita Levi Montalcini 4, Pieve Emanuele, 20090 Milan, Italy; 7Centro de Investigación Biomédica en Red de Enfermedades Hepáticas y Digestivas (CIBERehd), UGC Aparato Digestivo, Instituto de Investigación Biomédica de Málaga-IBIMA, Hospital Universitario Virgen de la Victoria, Universidad de Málaga, 29016 Málaga, Spain; andrade@uma.es

**Keywords:** inflammation, autoimmune hepatitis, autoimmunity, genetics, drug-induced liver injury

## Abstract

Drug-induced liver injury (DILI) is a challenging clinical event in medicine, particularly because of its ability to present with a variety of phenotypes including that of autoimmune hepatitis or other immune mediated liver injuries. Limited diagnostic and therapeutic tools are available, mostly because its pathogenesis has remained poorly understood for decades. The recent scientific and technological advancements in genomics and immunology are paving the way for a better understanding of the molecular aspects of DILI. This review provides an updated overview of the genetic predisposition and immunological mechanisms behind the pathogenesis of DILI and presents the state-of-the-art experimental models to study DILI at the pre-clinical level.

## 1. Introduction

Drug-induced Liver Injury (DILI) is defined by the presence of some degree of liver injury, commonly detected by the rise of liver-related enzymes in the blood, which can be causally related to a specific drug [1]. Estimates of incidence report around 15–20 cases per 100,000 individuals, and DILI accounts for half of the cases of acute liver failure in Western countries [1]. Registries such as the Spanish DILI Registry represent invaluable tools to study the epidemiology and disease course of DILI [2].

### 1.1. Types of DILI

Historically, DILI has been classified as direct (or intrinsic) and idiosyncratic. The direct type is characterized as dose-related, predictable and reproducible in animal models, since it is caused by chemical compounds intrinsically toxic for the liver. Examples of direct hepatotoxic agents are acetaminophen (at high doses), aspirin, and amiodarone [1]. On the contrary, the idiosyncratic type (iDILI) is less frequent, dose-unrelated, unpredictable and not reproducible in animal models, and it is associated with common, not intrinsically hepatotoxic drugs [1]. Examples of compounds that have often been associated with idiosyncratic liver toxicity are several antibiotics, such as amoxicillin-clavulanate, cephalosporins, fluoroquinolones and macrolides [1]. The onset is also different between the two types: while direct DILI takes some days to develop, iDILI has a quite variable onset, from days to weeks [3]. More recently, a third type of DILI, the indirect type, has been proposed: in terms of frequency, relation to the dose, predictability and reproducibility, it can be placed halfway between the other two types, has a slower onset (typically months) and it is mechanistically related to the pharmacodynamic properties of the compound [1]. For instance, there are growing reports of indirect DILI from immune-checkpoint inhibitors or rituximab in patients with chronic hepatitis B; in both cases, the damage is mostly mediated by the immune system [1]. Nonetheless, the presence of this third variant is still a matter of debate in the scientific community [4].

### 1.2. DILI vs. AIH: A Clinical Challenge

A frequent clinical challenge is represented by the differential diagnosis between DILI and Autoimmune Hepatitis (AIH), especially when AIH is seronegative (no detectable autoantibodies) or levels of serum Immunoglobulin G are normal. Histology could be of great value in these cases [5]. Severe interface hepatitis and the presence of rosettes or emperipolesis favor the diagnosis of AIH, whereas the presence of neutrophils in the portal tracts or intracellular cholestasis is more typical of DILI [5]. A marked decrease in serum transaminases after corticosteroid therapy has been recently proposed as ex juvantibus tool to discriminate DILI from AIH [6].

On the other side, DILI can manifest with autoimmune characteristics and drugs could represent the trigger of AIH [5]. In a landmark paper from the Netherlands evaluating the autoimmune features of patients with DILI taken from a prospective DILI registry, the authors found that about 40% of patients with DILI had increased Immunoglobulin G levels, together with high percentages (60–70%) of positivity for antibodies to nuclear antigen (ANA) and smooth muscle (SMA) but not for antibodies to soluble liver antigen (SLA) [7]. In contrast to classical AIH though, the investigators noticed that classical-risk HLA alleles associated with AIH were not present in these patients and titers of autoantibodies tended to decline over months [7]. Drug-induced AIH is reported to represent around 9% of all AIH cases [5] and its disease course is thought to be more favorable, with a higher probability of withdrawing immunosuppressive therapy without relapses [5]. A detailed description of clinical aspects of DILI is beyond the scope of this review; for clinical practice tips, see also guidelines on DILI from the European Association Study for the Liver [3].

## 2. Genetics of DILI

Host factors are supposed to play a key role in the development of DILI. Despite the wide variability in susceptibility and severity present in direct and indirect DILI, the pathogenesis of iDILI is even less clear. As a consequence, most genetic studies have focused on the idiosyncratic cases; iDILI is a multifaceted process, where host factors interplay together with environmental factors and the alleged drug [4]. The high degree of variance which characterizes iDILI speaks for a polygenic genetic architecture, where several risk variants are interconnected in a network where each provides a small effect size [8].

Before the Genome-Wide Association Studies (GWAS) era, many candidate gene studies were performed [9]. Most of them focused on genes involved in drug metabolism, since polymorphisms in their loci may have a profound impact and potentially account for toxicity at standard doses; a comprehensive list of the candidate genes stratified by their metabolic function (bioactivation, detoxification, clearance) can be found here [9].

Interestingly, no genetic variants identified in older candidate studies have been confirmed by GWAS. On the contrary, GWAS mostly pointed to the HLA region on chromosome 6, further highlighting the potential role of immunity in DILI.

Either the drug itself or its metabolites may behave as haptens and form neoantigens which bind to specific HLA proteins and ignite an inappropriate immune response. To build upon this concept, there is evidence that the same HLA allele may increase the risk for one drug and be protective for another one; unrelated drugs may share the same HLA allele too. Overall, these arguments reinforce the leading role of antigen presentation in the genesis of iDILI.

Regarding GWAS related to iDILI, most of them have examined the genetic predisposition to a single specific drug or class, whereas more recent studies have revealed some risk variants related to general predisposition independent of drug type [1,9,10].

Nicoletti et al. were the first to identify the rs114577328 Single Nucleotide Polymorphism (SNP) as a risk variant for unselected DILI cases involving different drugs in subjects of European ancestry [10]. Rs114577328 tags the HLA-A*33:01 allele, which is similar to the HLA-A*33:03 allele that has been associated to ticlopidine-induced DILI in Japanese individuals. Previous studies had identified examples of HLA alleles associated with structurally different compounds, such as DRB1*15:01 with amoxicillin-clavulanate and lumiracoxib and DRB1*07:01 with lapatinib and ximelagatran [11]. Nicoletti and colleagues speculate that their findings, together with the previously available evidence, support the role of adaptive immunity in DILI. The drug itself or its metabolites form adducts that bind HLA molecules and consequently activate T-cell response [10].

Recently, the same consortium also identified a genome-wide significant non-HLA risk allele, the rs2476601 SNP, on chromosome 1 [12]; the association was not constrained to a specific drug or pattern of injury. Further, the rs2476601 variant added to the risk associated with known HLA alleles, showing epistatic interaction [12]. The same SNP has been associated to increased risk of type 1 diabetes mellitus, rheumatoid arthritis, systemic lupus erythematosus, and several other autoimmune conditions [13]; it is considered one of the most important non-HLA allele for rheumatoid arthritis and juvenile idiopathic arthritis susceptibility [13].

The rs2476601 SNP tags the *lymphoid-specific protein tyrosine phosphatase non-receptor type 22* (*PTPN22*) gene which negatively controls several lymphocyte functions [14]. PTPN22 protein has an N-terminal catalytic domain, an interdomain, and a C-terminal binding domain; the latter includes four motifs, from P1 to P4. The rs2476601 polymorphism causes a R620W missense mutation in the P1 motif which prevents the interaction of the protein with c-Src kinase. How this structural disruption associates with functional autoimmunity is still under investigation. There is evidence that the rs2476601 SNP is associated with increased genesis of autoreactive B cell receptors, inhibition of T-cell receptor signaling and modification of T-cell adhesion [14]. Vang and colleagues have suggested that the PTPN22 R620W is a switch-of-function polymorphism which can operate as both a gain-of-function and a loss-of-function variant [14]. Indeed, its pathogenic role is also disease-specific, since the same variant is protective for Crohn’s disease and Behcet disease [13]. For a detailed review of PTPN22 structure, function and its role in autoimmunity see [13].

The rs2476601 allele frequency changes across populations: in Europe the highest frequency is found in northern and eastern Europe (>10%) and the lowest in southern Europe (2–3%) [13]. The allele is rare in Native American, African and Asian populations (<1%) [13]. In the aforementioned discovery study, rs2476601 variant showed an allele frequency of around 15% in Finnish subjects but <0.01% in East Asians; nevertheless, the effect size remained fairly consistent across ethnic groups [12].

Recently, the information on risk alleles derived from GWAS has been leveraged to develop a polygenic risk score (PRS) for DILI [15]. The authors of the study assessed the discriminative power of PRS in primary hepatocytes and stem-cell derived organoids, revealing that there is a shared DILI predisposition which is independent of chemical properties of each specific drug. They identified higher rates of inactivation of genes involved in mitochondria and translation in subjects with higher PRS values and pointed out that DILI susceptibility is due to several biological pathways in hepatocytes, including oxidative stress and unfolded protein response [15].

## 3. Immunology of DILI

### 3.1. Intrinsic DILI: Amplification of Liver Damage by Inflammation

Hepatocyte damage and dysfunctionality are directly caused by drugs in intrinsic DILI; yet, drug-induced innate immune response can amplify tissue damage. Indeed, innate immunity can be activated by damage-associated molecular patterns (DAMPs) released by drug-injured hepatocytes, resulting in inflammation in the absence of infectious agents, so-called sterile inflammation [16,17] (for definition of DAMPs and a brief outline on inflammation, see Box 1).
Box 1Inflammation.Inflammation is triggered by infections, toxic molecules,
and other agents inducing tissue injury. Tissue-resident innate immune cells
(such dendritic cells (DCs), macrophages, etc.) act as sentinels sensing the
environment via innate immunity receptors, including Toll-like receptors
(TLRs), RIG-like receptors (RLRs), NOD-like receptors (NLRs), etc., which
recognize either molecules associated to pathogens (pathogen-associated
molecular patterns, PAMPs, e.g., microbial nucleic acids, LPS, etc.), or
molecules released from damaged tissues (damage-associated molecular
patterns, DAMPs, e.g., intracellular proteins and/or metabolites, etc.) [18]. Innate receptor triggering activates
intracellular events culminating in the release of proinflammatory mediators,
cytokines and chemokines. These in turn recruit leukocytes from the blood
which, together with tissue-resident innate immune cells, contribute to
pathogen/toxic agent elimination via anti-microbial factors, phagocytosis,
etc. The innate immune system also plays an important role in tissue
remodeling and repair, by removing necrotic cells and cellular debris,
producing extracellular matrix-degrading enzymes and releasing growth
factors. Furthermore, some of the pleiotropic cytokines produced by
tissue-repairing macrophages have anti-inflammatory activity and modulate
adaptive immunity cells that switch off inflammation (i.e., regulatory T
cells, Tregs) [19,20,21,22].

DAMPs, such as high mobility group box-1 (HMGB1), keratin 18 (K18), and adenosine triphosphate (ATP), are currently studied as promising DILI biomarkers and therapeutic targets; this is the case for Acetaminophen (APAP) overdose, a very common cause of DILI in Western countries [23]. DAMPs activate liver-resident immune cells, including Kupffer cells (KCs, a population of liver-resident macrophages), NKT cells, γδ T cells and dendritic cells (DCs), thus triggering an inflammatory cascade involving neutrophils and monocyte recruitment from blood. Inflammatory cellular infiltrates are commonly observed by immunohistochemistry in liver biopsies from patients with acute DILI [24,25,26]. The contribution of different cell types to inflammation and tissue repair, and the underlying molecular mechanisms, have been further investigated in mouse models of DILI [27,28].

KCs are activated by DAMPs binding to TLRs and purigenic receptors [29]. The activation of KCs results in the production of proIL-1β and proIL-18, that are cleaved intracellularly by caspase-1, and released as mature IL-1β and IL-18, respectively [30]. IL-1β acts as mediator of neutrophil and monocyte recruitment, in conjunction with pro-inflammatory chemokines (e.g., CXCL1, CXCL2, CXCL8, CCL2, etc.), and amplifies the inflammatory process by activating infiltrating leukocytes [31]. IL-18 promotes Interferon gamma (IFN-γ) and Fas Ligand (FasL) expression, thus sustaining hepatic cell death and interfering with liver regeneration [32]. Furthermore, KCs can produce tumor necrosis factor alpha (TNF-α), which can kill hepatocytes and recruit inflammatory leukocytes in several types of liver injury [33,34] (Figure 1A). Additional studies have demonstrated that TNF-α is also a key factor for hepatocyte proliferation during liver regeneration in different conditions [35,36], including a mouse model of APAP-DILI [37]. It seems that low TNF-α concentrations promote proliferation, while high concentrations cause cell death, paving the way for new therapeutic approaches to DILI [38].

Natural Killer T (NKT) cells are abundant liver-resident innate lymphocytes whose role in liver inflammation has been demonstrated in several diseases [24,39,40]. NKT cells are considered likely players in DILI, as activated hepatic NKT cells are able to produce large amount of osteopontin and IL-17, two key cytokines attracting neutrophils [41,42,43], abundantly recruited into the liver in DILI [24]. Yet, NKT cells are not the only IL-17 producer in the liver. γδ T cells can produce a huge amount of this cytokine, and γδ T cell depletion—but not that of NKT cells—in a APAP-DILI murine model was associated with reduced IL-17A level, decreased neutrophil infiltration and attenuated liver damage [44]. Studies in mice genetically deficient in invariant NKT (iNKT) cells (Jα18^−/−^ and CD1d^−/−^) and in other mouse models have reported puzzling results about iNKT cell contribution to DILI. For example, in halothane-DILI, a pro-inflammatory role of iNKT cells has been suggested, as a greatly reduced neutrophil infiltration was observed in CD1d^−/−^ mice [45]. Similarly, in APAP-DILI, intrahepatic iNKT cell reduction or loss were associated to either ameliorated disease after treatment with glycosphingolipids and/or vitamin E [46], or increased glutathione levels and enhanced NAPQI (an APAP reactive metabolite) detoxification [47]. In contrast, another study showed exacerbated liver damage in both Jα18^−/−^ and CD1d^−/−^ mice after starvation, and this was associated with CYP2E1 up-regulation and enhanced formation of hepatic APAP-protein adducts, suggesting a protective role of iNKT cells [48]. Further studies are required to explain such discrepancies, possibly due to differences in mouse genetic background and/or DILI experimental models. In addition, DILI murine models only partially reflect the nuances of types and phenotypes of human DILI [1].

Neutrophils are a major component of liver infiltrate in DILI, as demonstrated by immunohistochemistry studies on liver biopsies [24]. Several chemokines, such as CXCL8, that binds to the neutrophil receptors CXCR1 and CXCR2, and CXCL1 and CXCL2, that both bind to CXCR2 receptor, rapidly attract neutrophils into the liver. Neutrophil recruitment is also mediated by β2 integrins expressed by neutrophils and adhesion molecules (such es ICAM-1 and VCAM-1) expressed by endothelial cells. β2 integrin-mediated interactions are also required for neutrophil/hepatocyte contact, that leads to the production of reactive oxygen species (ROS) by neutrophils, and hepatocytes damage and death [49,50]. Thus, neutrophils are implicated in hepatocyte killing and tissue injury in DILI, and play a similar role in alcoholic hepatitis [51,52,53]. Nevertheless, these cells can also contribute to liver regeneration and repair [36,54]. For example, neutrophil depletion by anti-Ly6G antibody treatment resulted in reduced hepatocyte proliferation and increased liver necrosis in mouse APAP-DILI [55].

Monocytes are attracted to the site of liver injury by the CCL2 chemokine, which binds to the chemokine receptor CCR2 expressed by these cells. Monocytes and liver macrophages contribute to the amplification of the inflammatory process, for example by producing proinflammatory cytokines, such as TNF-α, IL-6 and IL-1 [56,57]. Nevertheless, it should be noted that macrophages have heterogeneous functions, and their phenotype ranges from pro-inflammatory to anti-inflammatory/tissue-repair polarization, characterized by production of metalloproteinases (MMPs), fibronectin 1, VEGF-A, and anti-inflammatory cytokines, such as IL-10 [26,28,58,59].

### 3.2. Idiosyncratic DILI: Drug-Specific T Cell Response Triggered by Dendritic Cells

The cellular arm of adaptive immunity (i.e., T cells) plays a major role in iDILI, nevertheless the humoral arm of adaptive immunity (composed by B cells, plasma cells and antibodies) is also involved in some cases [1] (for a brief outline on T cell response, see Box 2). Supporting this concept, T-cell infiltrates are found in the liver of patients with iDILI, for example in liver biopsies of patients with either floxacillin- or amoxicillin-clavulanate-induced DILI [60,61]. T cells involved in DILI specifically recognize drug-peptides-MHC (in humans HLA) complexes.
Box 2T-cell response.The two main T-cell subsets are CD8+ T cells and
CD4+ T cells, that use their membrane T-Cell Receptor (TCR) to recognize
antigenic peptides in the context of class I and class II Major
Histocompatibility Complex (MHC-I and MHC-II) molecules, respectively. The
antigenic peptides derive from processing of cellular and extra-cellular
proteins, according to highly regulated mechanisms, so that proteins that are
synthetized inside the cell are normally presented in MHC-I (human HLA-I) by
all nucleated cells of the body, while proteins captured from the
extracellular space are usually presented in MHC-II (human HLA-II) by
specialized immune cell subsets. In healthy tissues, resting dendritic cells
(DCs) act as “immature” Antigen-Presenting Cells (APCs), continuously
presenting host protein-derived antigenic peptides (self-peptides) to T
cells, a mechanism that contributes to maintaining immune tolerance. In a
typical immune response, for example upon infection, DCs activated in the
infected tissue up-take and process pathogen-derived antigens, up-regulate
co-stimulatory molecules (e.g., CD80 and CD86, two B7 family members) and the
chemokine receptor CCR7, and migrate as “maturing” APCs to draining lymph
nodes (LNs) via lymphatic vessels, attracted by CCL19/21, two LN chemokines
recognized by CCR7 [62]. In LNs, “mature
DCs” prime resting naïve T cells by displaying pathogen-derived antigens in
the context of MHC and providing co-stimulatory signals and cytokines to the
T cells. Proliferation and differentiation of primed T cells generate a progeny
of effector T cells, that circulate all-over the body and are ready to exert
their function upon encounter of antigen-MHC complexes on the surface of
target cells, for example pathogen-infected cells. Effector CD8+ T cells kill
target cells via exocytosis of granules containing perforins and granzymes,
or via membrane interactions (FasL, TRAIL, etc.), while CD4+ T cells provide
help for activation of other immune cells, orchestrating diverse types of
responses. Indeed, many subtypes of effector CD4+ T cells have been
identified (e.g., Th1, Th2, Th17, Th22), each characterized by the production
of a specific set of cytokines. Furthermore, CD4+ T cells can mediate their
effector function through membrane molecules either providing co-stimulatory
signals, e.g., CD40L, OX40L, or inducing apoptosis via membrane interactions.
Effector T cells are short-lived, nevertheless a few memory T cells remain at
the end of primary immune response, and they can support strong secondary
responses upon antigen re-exposure.

The following steps are envisaged to explain iDILI pathogenesis. Drug-specific T-cell responses are triggered by liver DCs that take up drugs and/or their metabolites and display them in the form of drug-modified-peptide-HLA membrane molecules (Figure 1B). Thus, DCs bridge innate and adaptive immunity activation, acting as Antigen Presenting Cells (APCs). The current view is that sensitization, or priming, of naïve CD4^+^ and CD8^+^ T cells against drug-modified-peptide-HLA molecules can occur only if the multimolecular complexes are presented by fully activated, or “mature”, DCs in liver draining lymph nodes (LNs). This might occur if the drug itself, an abnormal drug metabolite, or the underlying disease requiring drug therapy, induce a mild liver-cell injury [1], with DAMPs release, resulting in full activation of liver-resident DCs. These cells change their membrane phenotype (up-regulating B7 molecules, the chemokine receptor CCR7, etc.) (Figure 1B), and via lymphatic vessels migrate to liver-draining LNs wherein they prime drug-specific naïve CD4+ and CD8+ T cells (Figure 1C). In support of this scenario, in vitro studies have shown that human hepatocytes treated with either flucloxacillin or nitroso-sulfamethoxazole release HMGB, that in turn triggers TNF-α, IL-6, and IL-1 production of by monocyte-derived DCs, and enhances the T-cell stimulatory capacity of these cells [63]. It remains to be determined whether cells other than liver DCs play a role in T cell priming in DILI, for example LN macrophages, or liver KCs, and endothelial cells, as shown in other conditions [64,65].

After extensive proliferation and differentiation in LNs, drug-specific effector CD4+ and CD8+ T cells are generated and released into the blood circulation. They can get recruited by increased chemokines into the liver [66], wherein they can be triggered upon recognition of drug-modified-peptide-HLA complexes on the surface of different types of liver cells (Figure 1D). All liver cells express MHC-I molecules and can become targets of cytotoxic CD8^+^ T cells, while normally only a few cells of hematopoietic origin express MHC-II molecules, such as hepatic DCs [67]. In stressed conditions or inflammation, e.g., in the presence of Interferons and/or TLR ligands, DCs up-regulate MHC-I and -II molecules, while KCs and liver endothelial sinusoidal cells become MHC-II+ [68]. Furthermore, MHC-II+ leukocytes recruited from peripheral blood infiltrate the liver. Thus, CD4+ T cells can be triggered to release cytokines by peptide-MHC-II complexes on the surface of several cell types in injured liver. Recruitment and activation of effector T cells into the liver can result in liver damage via different mechanisms, including production of diverse pro-inflammatory (IFN-γ, IL-17, etc.) and pro-allergic (IL-4, IL-5, IL-13, etc.) cytokines by CD4+ T cells, and cytotoxicity via either exocytosis of perforin- and granzyme-containing granules by CD8^+^ T cells, or FasL-mediated killing by both CD4+ and CD8+ T cells (Figure 1D).

The prevailing molecular explanation for T cell involvement in iDILI and in other drug-induced T-cell mediated reactions is that the causative drug and/or its metabolite are recognized by the TCR, after binding to self-molecules. There are two main possibilities: (i) non-covalent labile pharmacological interaction (p.i.) with immune receptors, such as self-peptide-HLA complexes, or even the TCR; (ii) covalent binding to self-proteins, so that the drug is a “hapten” and the protein a “carrier”, and the TCR recognizes a haptenized peptide in the context of HLA. The former mechanism does not require antigen processing, in contrast to the latter [69,70,71]. A well-known example of the “hapten” concept is the penicillin covalent binding to lysin residues, leading to presentation of penicilloyl peptides in the context of HLA to drug-specific T cells [69,71,72]. An alternative rare possibility is that the drug changes the HLA-binding site for the antigenic peptide, leading to a change in the repertoire of presented peptides [73]. The fact that individuals carrying certain HLA alleles have increased risk to develop DILI when treated with particular drugs can be explained by the preferential molecular association of the drug and/or its metabolites with certain peptide carrying-HLA alleles, for example flucloxacillin, also known as floxacillin, with HLA-B*57:01 [60], and amoxicillin with DRB1*15:01 and DQB1*06:02 [74]. Nevertheless, only a fraction of the drug-exposed individuals having the risk HLA allele develop DILI, suggesting that additional factors are required for disease development.

In vitro tests with peripheral blood T cells from DILI patients incubated with the causative drug or its metabolites can give information on the effector function of drug-specific T cells, thus providing insights on the mechanisms of T cell mediated liver damage [61]. For example, CD8+ T cell clones obtained from HLA-B*57:01^+^ patients with floxacillin-induced DILI exhibited dose-dependent proliferation, and production of IFN-γ, and of the cytotoxic molecules granzyme B, FasL, and perforin in response to floxacillin presented by autologous APCs. Clones were specific for floxacillin, but could also react to other β-lactam antibiotics, but not to abacavir [75]. CD4+ and CD8+ T cell clones obtained from patients with DILI elicited by the anti-microbial combination of amoxicillin and clavulanic acid were responsive in vitro to hapten-peptide-HLA complexes generated by either one or the other drug [61]. When stimulated in vitro in the presence of APCs, amoxicillin-specific CD4+ T cells were polyfunctional and secreted IFN-γ, IL-10, perforin and/or IL-17/IL-22 [61,74], while clavulanic acid-specific CD4+ T cells mostly produced IFN-γ [61]. Another example of drug-elicited T cell-effector function is the production of IFN-γ, IL-13, and granzyme B by CD4+ T cells from patients with tolvaptan-induced DILI [76].

Additional data support a pathogenetic role of T cells in iDILI. Anti-drug T cells express high levels of the chemokine receptors CCR2, CCR4, CCR9 and CXCR3, that can mediate cell migration and accumulation into the liver [61]. Furthermore, a missense mutation in the phosphatase protein tyrosine phosphatase, nonreceptor type 22 gene (PTPN22) [12] is a risk factor for iDILI, as discussed above. PTPN22 inhibits TCR signalling, and its missense mutation is also a risk factor for autoimmunity [13].

Adaptive immunity might contribute to DILI pathogenesis even in those cases in which an innate inflammatory infiltrate in the liver appears dominant. Thus, T cell priming and/or secondary stimulation can be induced by activated DCs and macrophages even in intrinsic DILI induced by direct drug toxicity. Furthermore, it should be noted that effector T cells often mediate liver injury through innate immunity mechanisms, that might dominate the pathological scenario, even though they are sustained by T cells. Thus, T-cell produced IL-17 can promote the recruitment of high numbers of neutrophils into the liver, that in turn cause tissue damage, or IL-4 and IL-5-producing T cells can sustain the so-called immunoallergic DILI with eosinophilia. Adding to this complexity, the adaptive immune system comprises a complex network of feedback and regulatory loops, thus some inhibitory cell subsets can dampen harmful responses (e.g., Treg cells), while others promote tissue repair (e.g., IL-22 producing T cells) [77,78]. Understanding the role of adaptive immunity in different types of DILI is especially relevant in consideration of immunological memory, the hallmark of adaptive immunity, that might underlie recurrent episodes upon re-exposure to the causative drug, and/or disease chronicity if the drug is not discontinued [1].

Finally, hypergammaglobulinemia and circulating autoantibodies (against CYP2E1 or liver endoplasmic reticulum proteins, among others), and liver infiltrating plasma cells are found in DILI with autoimmune features (AI-DILI), that can be induced by α-methyl DOPA, hydralazine, isoniazide, and other compounds [79,80]. Different mechanisms can underly AI-DILI, for example breaking of immune tolerance due to drug-induced cell damage with DAMP release, the formation of drug-protein adducts that stimulate innate immunity, which in turn activates adaptive responses against self-antigens [81]. The autoimmune reaction is usually self-limited, and disease resolves after drug withdrawal [1,82]. However, evolution into overt autoimmune hepatitis might occur in the presence of additional underlying factors [1,80].

### 3.3. DILI from Immune-Checkpoint Inhibitors: A Rising Clinical Issue

Recently, a novel phenotype of immune-mediated DILI has drawn much attention, following the sharp increase in the use of immune checkpoint inhibitors (ICI) in oncology and hematology. ICI represent a class of drugs that has noticeably improved the outcomes of multiple cancers, including, but not limited to, melanoma, lung cancer, head and neck cancer, renal cancer and haematological malignancies [83,84,85,86,87,88,89,90,91,92,93,94,95,96]. Due to their mechanism of action, ICI-related adverse events (AEs) are mostly immune-related AEs, and can affect any organ [97]. In most patients, these AEs are mild and reversible [98]. However, serious AEs occur in around 6–8% of patients, and could lead to fatal outcomes. Among the serious immune-related AEs, there are endocrinopathies, pneumonitis, colitis and hepatitis [99,100]. The incidence of liver toxicity induced by ICI may widely differ according to patients’ tumor type, type of ICI used, and different treatment combinations. A higher susceptibility to develop an ICI-induced liver toxicity has been observed in patients receiving immunotherapy combinations [101,102].

ICI-related DILI is often considered an immune-mediate hepatitis. ICI-induced liver toxicity differs from the other types of DILI because it is caused by an aberrant activation of immune system response, rather than by a direct hepatic damage or an idiosyncratic hepatotoxicity [1]. In other words, this indirect, ICI-induced, liver injury seems to depend on ICI inherent mechanism of action, and not on their intrinsic hepatotoxicity or immunogenicity [1].

ICI mechanism of action consists in “releasing the brakes” of the immune system, thus activating its response against tumor cells. Immune checkpoints are physiological molecules fundamental in the negative control of immune response, acting as minimizer or even suppressor of immune activity, in order to avoid potential tissue damage from aberrant inflammatory response [103]. Cancer evolution has brought some tumors to develop the ability to produce these molecules as a mechanism to escape from immune system control [104]. ICI include a variety of molecules targeting programmed cell death protein 1 (PD-1), such as pembrolizumab, nivolumab and cemiplimab, programmed cell death ligand 1 (PD-L1), such as atezolizumab, avelumab and durvalumab, and cytotoxic T-lymphocyte-associated protein 4 (CTLA4), like ipilimumab. PD-L1 is the binding partner of the PD-1 receptor, physiologically expressed on T cells. When PD-L1 binds to the PD-1 receptor, an inhibitory pathway is activated, aimed to maintain the self-tolerance. PD-1 or PD-L1 inhibitors act by preventing the interaction between the ligand and its receptor, thus inducing the persistent activation of immune response [105]. CTLA4 binds with CD80 or CD86 on the membrane of APC, acting as inhibitory signal.

Several open questions exist regarding ICI-induced liver toxicity. First, no specific biomarkers exist to identify liver toxicity induced by ICIs, and its diagnosis is made after exclusion of other possible causes, including disease progression to the liver, concomitant administration of hepatotoxic drugs or reactivation of silent viral infections (e.g., hepatitis B) [106]. The specific mechanisms underlying ICI-induced liver toxicity are not fully understood. Generally, a prevalence of inflammatory T lymphocytes can be observed, with a predominance of CD8+ cytotoxic cells, in patients receiving anti-CTLA4 antibodies, and with a more mixed CD8+/CD4+ infiltrate in those receiving anti PD-1 or anti PD-L1 [106]. Recent evidence shows activation of the peripheral monocyte cellular compartment together with increased activity of cytotoxic CD8+ T lymphocytes. These peripheral phenomena are reflected by liver inflammation dominated by co-localised CD8+ lymphocytes and CCR2+ macrophages [107]. Further research is needed to validate these preliminary, descriptive findings as biomarkers and to further understand the pathogenesis of ICI-induced hepatitis.

In addition, there is paucity of data about its specific pathological features. In ICI-induced liver toxicity, histology typically shows an acute hepatitis with necrosis more prevalent in the centrilobular areas; granulomas seem to prevail in patients treated with anti-CTLA-4 drugs [106]. Plasma cell infiltrates are not typical of ICI-induced hepatitis [106,108], in contrast to classical AIH, where they constitute the predominant cell type together with lymphocytes and accumulate in periportal areas [109]. Other hallmarks of AIH seem to be missing too, like autoantibodies or IgG elevations [106], even though evidence is still scarce and conflicting, with reports of about 50% of patients positive to ANA antibodies [106].

Moreover, whether development and evolution of ICI-related hepatitis differ between healthy or cirrhotic livers has not yet been elucidated, and it is unknown whether it may result in liver fibrosis [106]. The possible application of ICI in patients with cirrhosis and hepatocellular carcinoma makes the need to elucidate these aspects even more urgent [110].

Future studies are required in order to shed light on the pathophysiological mechanisms and molecular aspects of ICI-induced liver toxicity, in order to provide predictors of toxicity, as well as of resolution and recurrence, in case ICI therapy is resumed.

## 4. Experimental Methods

Experimental methods can be divided into high-throughput screening of toxicity of new drugs (called predictive models) and those devoted to the study of the pathogenesis of DILI (in vitro and in vivo models).

### 4.1. Predictive Models

Since DILI represents the major cause of medications withdrawal from the market [111] and of acute liver failure events around the world [112], the development of methods possibly revealing hepatotoxic effects of drug candidates during the first phases of production are of paramount importance. Predictive models can improve drug safety as well as reduce the time and cost of drug development [113].

In addition to the inter-individual biodiversity, the cellular mechanisms of DILI hepatotoxicity depend on the culprit drug [114]. As a consequence, to determine the physiochemical properties of the compound is crucial. Factors associated with the risk of DILI are lipophilicity, number of hydrogen-bond donors and acceptors, polar surface area as well as pharmacokinetic parameters such as high systemic and daily drug exposure [115,116,117].

Planning in vitro/in vivo studies implies considering all the pharmacokinetic phases: absorption, distribution, metabolism and excretion, together with the target drug exposure (concentration). In this context, “chemical structure-based in silico models”, “toxicogenomics-based models” and “in vitro-based models” have been developed [118].

Chemical structure-based in silico models are based on the assumption that similar chemical structures show similar mechanisms of toxicity [113]. This method provides the opportunity to screen a large number of compounds without the need to synthesise them. Specific chemical substructures or motifs are statistically associated with elevated risk of toxicity by commercial software [119]. The system generates alerts that need expert validation to avoid inaccurate predictions. To avert expert opinion and make the approach faster, supervised machine learning methods have been progressively used (e.g., QSAR-based models) [120].

Toxicogenomics-based models use microarray-based technology to assess alterations in gene expression-profiles induced by drugs. Yet, the predictive power of this model is not as effective as envisioned; the alteration in the gene expression is not necessarily relevant and associated to DILI [113].

In vitro-based models have been widely used in preclinical testing. Unfortunately, the conventional assays testing a single endpoint show very poor predictive power for DILI, showing a sensitivity of less than 25%. To overcome this limitation, one can use a panel of characterized assays in parallel, studying simultaneously multiple endpoints in live cells: to name the most important ones, steatosis, ROS production, phospholipidosis, mitochondria membrane integrity, oxidative stress, intracellular calcium homeostasis, inhibition of biliary efflux, cholestasis, lysosomal impairment [121]. In addition, the predictive power is higher when the assays target early steps in the process of cell injury, before cell death stages [122].

To summarise, current predictive methods are still inaccurate and there is an urgent need for improvement [122]. Overall, a valid predictive model should integrate chemical structure, cellular endpoints, multiple data sources and toxicogenomics data [113,123]. Artificial intelligence methods are expected to provide an important advancement to this end.

### 4.2. In Vitro Models

In addition to their role as predictive models, in vitro models are essential to study the molecular pathways of drug toxicity and the consequent pathogenic mechanisms of DILI. The complex network of interactions among genetic, environmental, immune and metabolic factors behind DILI implies that its pathophysiological mechanisms cannot be recapitulated in a single study model [116].

Primary human hepatocyte cultures are considered the gold standard to elucidate the pathogenesis of DILI. They reflect the in vivo metabolism, by retaining the expression of both Phases I and II enzymes, and toxicity. Nonetheless, their major limitations are (i) poor availability, (ii) reliance on biological features of the donor and (iii) the rapid loss of phenotypic characteristics and viability after few passages [118,124].

On the other side, human hepatic cell lines, either derived from hepatocellular carcinoma or immortalized, have stable phenotype, unlimited propagation potential and availability. Yet, most of them lack drug metabolizing enzymes and express a reduced number of hydrophilic drug transporters when compared to primary hepatocytes cultures. Moreover, they do not present normal cellular toxic responses since derive from liver cancer. Among the most used cell lines (Huh7, SK-Hep-1, HepaRG, Hep3B, HepG2, HiHeps, L-02 et al.), HepaRG is a promising predictive in vitro model and it is composed by a mixture of biliary-like cells and hepatocytes [125,126]. HepaRG can proliferate and differentiate to form colonies surrounded by epithelial cells expressing metabolizing enzymes and drug transporters [127,128].

The use of human induced pluripotent stem cell for derivation of hepatocyte-like cells represents another option despite remaining suboptimal, because cells are not fully metabolically competent and show very low levels of activity of the metabolizing enzymes [129,130]. Some potential improvements of this model have been proposed [131]. For example, Ware et al. [132] used a micropatterned coculture, a model housing stem cell-derived hepatocytes and murine embryonic fibroblasts in collagen-treated tissue culture array. This model exhibits slightly higher predictive power when compared to primary human hepatocytes (65% vs. 70%). The non-parenchymal component of the liver (endothelial, stellate, biliary and Kupffer cells) has been shown to be critical to recreate liver/hepatocytes function and liver mechanisms of injury [133,134,135]. Additionally, an environment that reflects both the in vivo organ scaffolds (such as extracellular matrix) and mechanical signals (like fluid flow) may drastically improve cited models [136,137]. The organization into a 3D tissue structure of the co-culture system has certainly moved the field forward in the recent years. The main advantages are the evaluation of DILI by a higher expression level of genes involved in the metabolism of chemicals as well together with the higher similarity with the liver tissue architecture [124,138,139,140].

Another significant example is derived from Bell and colleagues, who generated spheroids from primary human cryopreserved hepatocytes and non-parenchymal cells and compared them to the 2D cell models [141]. The spheroids remain phenotypically stable up to five weeks in culture, retaining metabolic function, viability and morphology. Interestingly, this model can predict DILI of the chemical fialuridine, which goes undetected using previous in vitro systems [142]. In addition, the 3D spheroid model maintains the pharmacokinetic differences among donors, enabling the study of the human variability in DILI [143]. Overall, the spheroid system exceeds all the predictive power of the previously published in vitro assays. An important caveat is that vascular structures and cholangiocytes cannot be generated in the 3D spheroids and the model does not include a physiological blood flow, which hampers both detoxification capacity and oxygen and chemical gradients [124,144].

To overcome this limitation, perfusion culture systems (starting from 3D models) have been recently developed. In some cases, even vascular- and bile canaliculi-like structures have been observed [145,146]. However, several aspects need to be improved. Circulating media are replaced with new and fresh ones, causing the loss of signals from soluble factors released by the hepatic tissue itself. Furthermore, although these systems allow for intercellular interactions and are responsive to inflammatory stimuli, they still have not solved the problem of the lack of adaptive immunity [116].

Hence, animal testing seems to be a necessary step in iDILI studies, especially where the study of the adaptive immunity is concerned. However, numerous interspecies differences of liver pathways are present, suggesting that predictive human-based in vitro models can be preferable.

To this end, the convergence of several recently developed microscale technologies (such as engineered biomaterials, microfluidic devices, physiological media, patient-derived primary hepatocytes, synthetic biology etc.) has to be used for the development of increasingly efficient “3D microphysiology systems” (miniaturized functional liver units constructed with multiple cell types under a variety of mechanical and biochemical cues) for the study of DILI [116,147,148].

### 4.3. In Vivo Models

The identification of suitable animal models for DILI has probably not reached its goal yet. Although intrinsic DILI models could be regarded as straightforward, since it is usually enough to treat the animals with large doses to induce liver damage, one should keep in mind that each animal could require a different dosage because of the intrinsic variability of liver metabolic processes across species.

The best studied model of intrinsic DILI is the one due to acetaminophen overdose. The primum movens is the conversion of APAP to NAPQI, process catalyzed by cytotochrome P450 enzymes; this metabolite binds to glutathione and proteins, including mitochondrial ones, affecting also cellular respiration. Due to the metabolic passages involved in damage generation, it is easily understandable that different animal species, and even different strains, can present a variable liver damage after APAP overdose. The best animal model to study liver damage after APAP administration is represented by mice, in which toxicity is induced with doses ≥150 mg/kg, similarly to what observed in humans [110]. Conversely, rats are more resistant to APAP damage, possibly due to a higher scavenger enzyme activity and thus to reduced oxidative stress [110]. In fact, induction of liver damage in rats can be obtained if APAP administration is preceded by a strong reduction of anti-oxidant capacity, either through a reduction in GSH production [110] or in SOD activity [149]. It must be noted, however, that there are also differences among mice strains, mostly due to the presence of spontaneous mutations or polymorphisms in the genome that affect either the antioxidant capacity or their ability to respond to endotoxins [150,151].

In intrinsic DILI there is a clear involvement of innate immunity, and the liver damage induced by APAP can also be modulated by neutrophils and macrophages, as previously explained. Neutropenia induced by the use of anti-neutrophil antibodies before APAP overdose can reduce hepatotoxicity [152], but an important role seems also to be played by Kupffer cell activation, since a depletion of this liver component significantly reduced liver damage in APAP overdose in mice [153].

The identification of a “perfect” animal model for iDILI remains even more elusive. The development of iDILI involves several mechanisms, i.e., induction of inflammation, loss of immune tolerance and possibly mitochondrial dysfunction. For this reason, researchers tried to develop animal models of iDILI following one of these possible triggering pathways.

To resemble the inflammatory environment underlying iDILI, animals were treated with bacterial lipopolysaccharides (LPS) to induce inflammation. However, several researchers used different schemes of LPS treatment, differing especially for administration timeframe, i.e., either before, at the same time or after the drug involved in iDILI development [154,155,156]. In addition to the possible variables due to the different treatments, this approach presents a flaw related to the immune response involved in iDILI pathogenesis. In fact, LPS mainly triggers the innate immunity response which, as mentioned, has a pivotal role in intrinsic DILI, whereas in iDILI there is a major involvement of adaptive immunity.

A completely different approach was taken by other researchers who were able to develop an iDILI mouse model modulating inhibitory receptors controlling adaptive immune response and immune tolerance. This model (also named “Uetrecht-Pohl”) was conceived starting from the idea that few patients develop iDILI whereas most of the individuals taking the same drug, after an initial mild damage, develop immunotolerance. This mouse model has thus been developed altering the function of two different genes, namely PD-1 and CTLA4 [157]. PD-1 has a double role in controlling T-cell response, since it is an immune-inhibitory receptor expressed in activated T cells, involved in the regulation of effector CD8+ T cells, but also able to promote the differentiation of CD4+ T cells into Tregs. In addition, CTLA-4 controls the T-cell mediated immune response both interfering with the activation of CD4+ cells after antigen presentation and by reducing IL-2 expression. This mouse model is thus based on PD-1^−/−^ mouse which is also treated with anti-CTLA4 antibodies, and it has been used to evaluate the development of iDILI after treatment with various drugs and food supplements [157,158,159]. Although this model might resemble the delayed onset and some other elements of the pathogenetic process happening in humans, it still lacks the progression towards severe liver damage and liver failure [160].

As regards the mitochondrial disfunction, Ong et al., reported that mice with a reduced level of Sod2 developed mild liver injury after administration of troglitazone [161], but these data were not further confirmed and, on the contrary, experiments performed in PD1^−/−^ anti-CTLA4-treated mice do not seem to support such an important role of mitochondrial damage in iDILI [160].

To conclude, current in vitro and in vivo model systems are still not adequate enough to understand the pathogenic mechanisms of this complex condition, particularly for the idiosyncratic type [118].

## 5. Conclusions

Both innate and adaptive immunity have a clear and pivotal role in intrinsic DILI and iDILI, and genetic factors of the host regulate individual susceptibility in iDILI. Despite the advancement in the understanding of the immunogenetics of iDILI, we highlighted the complexity of studying host predisposition and the current limitations of available experimental DILI models.

DILI represents a great burden in terms of morbidity and mortality and a significant obstacle for drug development. There is urgent need for better experimental models that recapitulate the inflammatory and immunological abnormalities observed in the different types of DILI, together with accurate biomarkers to test in in vivo models. Hybrid approaches leveraging new biotechnological techniques together with novel computational methods should foster future research efforts aimed at improving our understanding of molecular and cellular mechanisms of DILI.
Figure 1(**A**–**D**) Innate and adaptive immune responses in DILI.
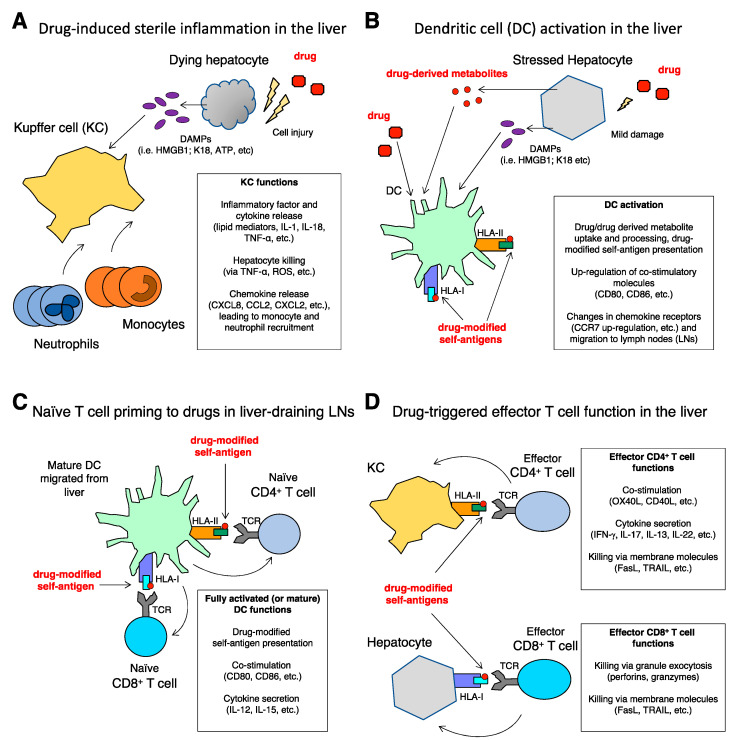


Examples of immune-mediated mechanisms are illustrated (see text for more details).

(A) Drug-induced sterile inflammation in the liver. Drug-mediated injury of hepatocytes is followed by the release of DAMPs that are sensed by Kupfer cells (KCs) which, in turn, start the inflammatory process characterized by the recruitment of neutrophils and monocytes from the blood. Both lipid mediators and cytokines sustain the inflammation. Hepatocyte killing by cells of the innate immunity contributes to liver damage and amplifies the inflammatory loop. A proto-typical example of drug-induced sterile inflammation is that occurring in APAP-DILI; in this scenario, inflammation can be inhibited by neutralization of HMGB1, one of the DAMPs released by hepatocytes [14,21,27].

(B) Dendritic cell (DC) activation in the liver. DAMPs released from injured hepatocytes activate DCs, which up-regulate co-stimulatory molecules, take up drugs and/or their metabolites, and present drug-modified self-antigens in the context of HLA-I and HLA-II. Activated DCs modulate chemokine receptors expression and migrate to liver-draining lymph nodes (LNs) wherein they can induce a drug-specific T cell response. In vitro results support the depicted mechanisms, for example by showing monocyte-derived DC activation after incubation with DAMPs released by primary human hepatocytes exposed to floxacillin, amoxicillin, and isoniazid [61]. Notably, floxacillin and amoxicillin are typical examples of drugs able to form adducts, that in turn can act as drug-modified self-antigens stimulating T cells [58,72].

(C) Naïve T cell priming to drugs in liver-draining LNs. Naïve T cells are primed in the LNs by fully activated or mature DCs, that present drug-modified antigenic peptides in the context of HLA-I and HLA-II to CD8+ and CD4+ T cells, respectively, and provide co-stimulatory signals and cytokines. CD4+ and CD8+ T cells recognizing drug-modified antigenic peptides proliferate, differentiate into effector and memory T cells, and circulate into the blood. The proposed scheme of T-cell priming is in agreement with in vitro experiments showing enhanced T-cell priming to nitroso-sulfamethoxazole, a sulfamethoxazole derivative, after exposure of monocyte-derived DCs to HMGB1 isoform A, one of the most investigated DAMPs [61].

(D) Drug-triggered effector T cell function in the liver. Effector CD8+ and CD4+ T cells migrate from the blood into the liver, wherein they are triggered by different types of cells that present drug-modified antigenic peptides in the context of HLA-I and HLA-II, respectively. All cells in the liver are HLA-I+ (e.g., hepatocytes), while HLA-II+ cells include activated KCs (shown as an example), endothelial cells, liver infiltrating leukocytes, etc. Effector CD4+ T cells orchestrate immune response by up-regulating membrane costimulatory molecules and producing cytokines. They can also kill target cells via membrane molecules. Effector CD8+ T cells are mostly cytotoxic, and kill target cells via either granule exocytosis or membrane molecules. The depicted mechanisms are supported by examination of liver infiltrates from patients with floxacillin-induced DILI [58], and by in vitro studies of effector functions by CD4+ and CD8+ T cell clones obtained from patients with DILI induced by amoxicillin, clavulanic acid and tolvaptan [59,72,74].

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
