# Peer review of "Immune-Mediated Drug-Induced Liver Injury: Immunogenetics and Experimental Models"

_ijms, 2021, doi:10.3390/ijms22094557_

Round 1
Reviewer 1 Report
Reviewer’s comments
This review article primarily focused on the current trends in drug-induced liver injury. The article seems to be quite impressive. It provides us many novel evidences in patients with drug-induced liver injury or its experimental animal models. I greatly appreciate your efforts. However, several revisions are required for publication. Please refer to the comments shown below.
Major
#1. At the beginning of Introduction, the authors should describe the classification of drug-induced liver injury; direct hepatotoxicity, idiosyncratic hepatotoxicity and indirect hepatotoxicity. In addition, they should list up the most commonly implicated agents in each category.
#2. The authors revealed that overdose of acetaminophen caused drug-induced sterile inflammation in the liver (Fig.1A). But, they did not describe which drug caused DC activation in the liver (Fig.1B). Likewise, they should also explain Fig.1C and Fig.1D in detail.
#3. The authors should describe the main points which discriminate DILI with autoimmune features from overt autoimmune hepatitis more clearly. Moreover, they should mention which types of autoantibodies (antinuclear antibodies? smooth muscle antibodies? or antibodies to liver kidney microsome?) are present in sera of patients with DILI with autoimmune features.
#4. The statement, “Immune checkpoint inhibitors may trigger autoimmune diseases such as autoimmune hepatitis” should be added. Moreover, histological characteristics caused by immune checkpoint inhibitor-related DILI should be described in more detail.
Minor
#1. Many abbreviation are used in the article. A list for the abbreviations should be made. In addition, PAMPs, LPS, TLR, TRAIL and NO should be spelled out.
#2. “Tumor Necrosis Factor-alpha” should be corrected to “tumor necrosis factor-alpha” (Line 190).
#3. “Plasmacells” should be corrected to “plasma cells” (Line 378).
Reviewer 2 Report
This is a very comprehensive review of Drug Induced Liver Injury. It aims to summarise the current knowledge of the different types of DILI, the genetics and immunology that underlie them and to evaluate the currently available pre-clinical research models.
This review addresses the genetics and immunology underlying the different types of DILI in a lot of detail. The figures offer a strong illustration of the immune responses. A drawback of having so much detail throughout is that it can be difficult for the reader to identify the most important ideas - for example, are the supplementary boxes (particularly box 1) required?
Specific comments:
- In line 99, I think that the reference number is incorrect (possibly should be 9 not 7).
- In lines 462-463, the claims should be supported with a reference
- in line 462, I would recommend specifying the withdrawal of medications from market to remove any confusion with drug withdrawal
Round 2
Reviewer 1 Report
The revised review article seems to be much better. The authors responded to the reviewers' comments very well. But, the article requires minor revision.
#1.”Drug-induced liver injury " should be added as a key word.
#2. "Cancer chemotherapy" is not a drug (line 48). The authors should correct it.
Author Response
We thank the reviewer for the positive comments.
1: we added Drug-induced liver injury as additional keyword
2: we removed cancer chemotherapy from the list. We believe that the list of drugs used in cancer chemotherapy is too long and would not be fair to mention one over another.